# CXCL14 Attenuates Triple-Negative Breast Cancer Progression by Regulating Immune Profiles of the Tumor Microenvironment in a T Cell-Dependent Manner

**DOI:** 10.3390/ijms23169314

**Published:** 2022-08-18

**Authors:** Carla Gibbs, Jae Young So, Abdul Ahad, Aleksandra M. Michalowski, Deok-Soo Son, Yang Li

**Affiliations:** 1Laboratory of Cancer Biology and Genetics, Center for Cancer Research, National Cancer Institute, Bethesda, MD 20892, USA; 2Department of Biochemistry, Cancer Biology, Neuroscience and Pharmacology, Meharry Medical College, Nashville, TN 37208, USA

**Keywords:** triple-negative breast cancer, tumor microenvironment, chemokines, CXCL14, immune profiles

## Abstract

Triple-negative breast cancer (TNBC) is aggressive and has a poor overall survival due to a lack of therapeutic targets compared to other subtypes. Chemokine signature revealed that TNBC had low levels of CXCL14, an orphan homeostatic chemokine to regulate the immune network. Here, we investigated if CXCL14 plays a critical role in TNBC progression, focusing on survival rates, tumor growth and metastasis, and immune profiles in the tumor microenvironment. Analysis of human breast-cancer datasets showed that low *CXCL14* expression levels were associated with poor survival rates in patients with breast cancer, particularly for TNBC subtypes. Overexpression of CXCL14 in TNBC 4T1 orthotopic mouse model significantly reduced tumor weights and inhibited lung metastasis. Furthermore, the CXCL14 overexpression altered immune profiles in the tumor microenvironment as follows: decreased F4/80+ macrophages and CD4+CD25+ Treg cells, and increased CD8+T cells in primary tumors; decreased Ly6C+ myeloid cells and CD4+CD25+ Treg cells and increased CD4+ and CD8+T cells in lung metastatic tumors. CXCL14-induced reduction of tumor growth and metastasis was diminished in T cell-deficient nude mice. Taken together, our data demonstrate that CXCL14 inhibits TNBC progression through altering immune profiles in the tumor microenvironment and it is mediated in a T cell-dependent manner. Thus, CXCL14 could be used as a biomarker for prognosis.

## 1. Introduction

Based on the expression of estrogen receptor (ER), progesterone receptor (PR), and human epidermal receptor 2 (HER2), breast cancer is subdivided into the following groups: luminal A (ER+/PR+/HER2-), luminal B (ER+/PR+/HER2+), HER2 enriched (ER-/PR-/HER2+), and basal-like, including triple-negative breast cancer (TNBC), which lacks ER, PR, and HER2. TNBC is the most aggressive form of breast cancer and has poor prognosis because of lack of therapeutic targets like ER, PR, and HER2 [1,2]. Although overall breast-cancer incidence has decreased, African-American (AA) women continue to have the highest mortality rate of TNBC cases (e.g., 42% higher compared to European American women when adjusted for incidence rate) [3,4], showing health disparities that need to be addressed. Furthermore, features of TNBC link to comorbidity with obesity, BRCA1 amplification, and brain metastasis [5,6,7]. As a result of the inherent heterogeneous profile of TNBC patients, TNBC can be subdivided into six subtypes: basal-like (BL1 and BL2), mesenchymal-like (ML), mesenchymal stem-like (MSL), immunomodulatory (IM), and luminal androgen receptor (LAR) [8]. Because the LAR subtype is rich in androgen receptor, it obtains a clinical benefit from anti-androgen therapies [9]. While TNBC patients can benefit from front-line chemotherapy such as anthracyclines, some patients experience relapse and resistance, which remains a key challenge of TNBC [10]. TNBC resistance is followed by aggressive metastasis, which leads to a causal factor for cancer death [11,12]. Understanding of TNBC biology is still limited due to high heterogeneity and constant evolution and chemoresistance. In particular, the TNBC microenvironment remains an attractive niche for investigating novel biomarkers and treatment options [13].

Chemokines are a family of chemo-attractant cytokines consisting of four subgroups based on the number of amino acids between the initial cysteine (C) motifs as follows: C (XCL1-2), CC (CCL1-28), CXC (CXCL1-17) and CX3C (CX3CL1). The main function of chemokines is to regulate cell migration by binding to specific cell surface G protein coupled receptors in development, physiology, and immune responses [14]. In our previous study, chemokine signature revealed that TNBC cells highly expressed proinflammatory chemokines CXCL1 and CXCL8 compared to non-TNBC cells [15]. Interestingly, BL subtype, including TNBC, has the lowest levels of CXCL14 compared to other subtypes [15], indicating a critical role of CXCL14 in TNBC. Although CXCL14 as an orphan chemokine is poorly understood, most recently, CXCL14 has been posited as an immune mediator in many cancer types [16]. Other studies have identified the role of CXCL14 in the innate immune response from its inhibition of proinflammatory cytokines in acute kidney injury and the regulation of T cells in stroke [17,18,19,20,21], indicating a pivotal role of CXCL14 as an anti-inflammatory and immune mediator. Elevated levels of CXCL14 are shown to be correlated with improved survival in breast-cancer patients based on histological staining [22]. These results led us to question whether CXCL14 could promote an anti-tumor environment in TNBC. Consequently, this study was designed to investigate if CXCL14 plays a critical role in TNBC progression, by analyzing patient survival rates from human datasets, tumor growth, and metastasis from an in vivo animal model, and immune profiles in the tumor microenvironment of primary and metastatic sites using flow cytometry. We demonstrate that CXCL14 inhibits tumor growth in primary and metastatic sites in a T cell-dependent manner through altering immune cell infiltration in the tumor microenvironment. Our work suggests that CXCL14 could be used as a biomarker for prognosis.

## 2. Results

### 2.1. High Levels of CXCL14 Correlated with Better Survivals in Patients with Breast Cancer

To investigate the clinical relevance of CXCL14 in breast cancer, we examined the correlation between *CXCL14* expression and survival of breast cancer patients using publicly available human breast-cancer datasets (Figure 1). High levels of *CXCL14* showed better overall survival (OS), recurrence-free survival (RFS), and distant metastasis-free survival (DMFS) in patients with breast cancer (Figure 1), indicating CXCL14 as a prognostic marker for favorable clinical outcomes in breast cancer.

### 2.2. Comparison of CXCL14 Expression Levels and CXCL14-Contributed Survivals in Breast Cancer Subtypes

The correlation between *CXCL14* expression and clinical features was further investigated among different breast-cancer subtypes. Basal-like subtype showed the lowest expression levels of *CXCL14* in multiple datasets including TCGA, METABRIC and Yau (Figure 2A). We also examined the relationship between *CXCL14* and OS in different breast-cancer subtypes. Both the KM plotter and the METABRIC datasets showed significantly decreased OS with low *CXCL14* expression in LumA and Basal-like subtypes (Figure 2B,C). Of great interest, this *CXCL14* correlation was not observed in LumB and HER2 subtypes, but with a weak significance (*p* = 0.04) for HER2 subtype in METABRIC dataset (Figure 2B,C).

As aggressive breast cancers have been diagnosed in younger women, we then evaluated the association between *CXCL14* expression and age. In a cohort of 51 breast cancer patients GSE18864 and in the TCGA breast cancer dataset, *CXCL14* expression was overall higher in older breast-cancer patients than younger patients post chemotherapy (Appendix A). The TCGA data of protein expression with a cohort of 410 patients across ages 30 through 90 also showed overall higher CXCL14 protein level in older patients (Appendix A). Compared to breast normal tissues, tumoral tissues presented lower *CXCL14* expression levels and metastatic tissues showed the lowest levels (Appendix A). Patients with high lymph-node metastasis showed significantly decreased *CXCL14* expression levels (Appendix A). Together, these data demonstrate the inhibitory role of CXCL14 on breast-cancer progression.

### 2.3. Overexpression of CXCL14 Decreased Breast Tumor Growth and Metastasis

To investigate biological functions of CXCL14, we generated CXCL14 overexpressing 4T1 cell clones using several promoters driving the CXCL14 expression (Appendix A). Among over 50 clones, we selected the 4T1-Fer-C1 clone, which showed the highest expression levels of CXCL14 by ELISA (Figure 3A). We then performed in vivo animal experiments as shown in the experimental design, and tumor phenotype was evaluated at day 12, 18, and 28 after the cancer cell injection (Figure 3B). The tumor weights in mice bearing CXCL14 overexpression cells were significantly lower than those in mice bearing control cells at all three time points (Figure 3C). The number of lung metastases was significantly decreased in mice bearing CXCL14 overexpression compared to mice bearing control cells at day 28, the end point of the experiment (Figure 3D). Impressively, in the CXCL14 overexpression group, 3 out of 7 mice showed no metastases within the lung (Figure 3D). These results demonstrated the inhibitory role of CXCL14 on tumor growth and metastasis of breast cancer in vivo.

### 2.4. CXCL14 Modulates Immune Profiles in the Tumor Microenvironment in Primary Tumors and Metastatic Sites

We further investigated the effects of CXCL14 on the immune-cell profiles in the tumor microenvironment using FACS analysis. The surface markers of immune cells in myeloid and lymphoid panels were listed (Appendix A) and immune cells were profiled with the myeloid cell-specific gating (Appendix A) and lymphoid cell-specific gating (Appendix A). As for the myeloid subsets, CXCL14 overexpression had no change in primary tumoral F4/80+ macrophages but decreased metastatic pulmonary F4/80+ macrophages at day 28 (Figure 4A). The analysis of F4/80+ macrophages in live cells also showed similar results (Appendix A). Conversely, CXCL14 overexpression decreased in primary tumoral Ly6C+ myeloid cells at day 18 and 28, but had no change in metastatic pulmonary Ly6C+ myeloid cells (Figure 4A). As for lymphoid cells, CXCL14 overexpression reduced both primary tumoral and metastatic pulmonary CD3+ T cells at day 28 (Figure 4B,C). In further analysis, this CD3+ T cell decrease is mostly due to reduced number of CD4+ CD25+ T regulatory cells (Treg), which was observed at day 28 in primary tumors and at day 12 and 28 in metastatic lungs (Figure 4B,C). Conversely, CXCL14 overexpression increased the number of CD8+ T cells at day 28 in primary tumors and at day 12 and 18 in metastatic lungs compared to control (Figure 4B,C). CD4+ T cells had no change in primary tumors between control and CXCL14 overexpression (Figure 4B). Lung metastatic sites showed fluctuations of CD4+ T cell infiltrates as follows: decreased infiltration at day 18 and increased infiltration at day 28 (Figure 4C). There were no substantial changes in CD45+ cells by CXCL14 overexpression at day 18 and 28, which are the time points at which we found the most differences in immune-cell compositions (Appendix A). In our data, we found a significant decrease in B cells in the overexpression compared to the control at Day 28. (Appendix A). There were no changes in NK cells by CXCL14 overexpression (Appendix A). These results indicate that CXCL14 plays a critical role in immune cell infiltration in the tumor microenvironment of both primary tumors and metastatic sites.

### 2.5. CXCL14 Attenuates Tumor Growth and Metastasis in a T Cell-Dependent Manner

CXCL14-induced decrease in Treg cells and increase in CD8+ T cells in primary tumors and metastatic lungs likely contributed to the tumor-inhibitory effects of CXCL14. Therefore, we used T cell-deficient athymic nude mice to determine whether T cells play a causal-effect role. CXCL14 overexpression did not change tumor weight in nude mice (Figure 5A, left panel), while it decreased tumor weight in wild-type BALB/c mice (Figure 3C), resulting in an increased tumor weight in nude mice compared with WT Balb/c mice (Figure 5A, right panel). Furthermore, CXCL14 overexpression did not change the number of lung metastases in nude mice (Figure 5B, left panel), while it decreased metastasis in wild-type BALB/c mice (Figure 3D), resulting in increased metastatic nodules in nude mice compared with WT Balb/c mice (Figure 5B, right panel). Together with our data from Figure 3, these results suggest that the T cells plays a critical role in the CXCL14-induced inhibition of tumor growth and metastasis in breast cancer.

### 2.6. Correlation of CXCL14 Expression with Immune Modulators in Breast Cancer Subtypes

One of the main functions of chemokines is to regulate immune cell migration and modulate the immune microenvironment. We, next, investigated the correlation between *CXCL14* expression and immune profiles in human breast cancer. The immune cell subsets were deconvoluted from the TCGA dataset by using XCELL algorithm. When all tumor samples analyzed without stratifying subtypes, *CXCL14* expression showed negative correlation with infiltration of myeloid derived suppressive cell (MDSC), CD4+ Th1 and CD4+ Th2 cells (Figure 6A). This negative correlation between *CXCL14* and MDSC infiltration was also found in Basal-like and HER2+ subtypes (Figure 6A). Interestingly *CXCL14* expression showed positive correlation with CD8 T cell only in Basal-like subtype (Figure 6A).

TIDE (Tumor Immune Dysfunction and Exclusion) analysis, which estimates the cytotoxic T lymphocyte level (CTL score) in tumor samples, also demonstrated a positive correlation between *CXCL14* expression and CTL score in TNBC (Figure 6B). In co-expression analysis between *CXCL14* and CD8 T cell functional genes, *CXCL14* expression was found to be positively correlated with *CD8A* and *CD8B* in TNBC (Figure 6C). Interestingly, the genes associated with cytotoxic T cell activity, such as *CXCR3*, *GZMA* and *FASLG*, showed positive correlation with *CXCL14*, while *FOXP3*, which is a key functional maker for Treg, showed a negative correlation with *CXCL14* (Figure 6C). In addition, in single-cell RNA-seq data of human breast cancer (GSE176078), we found a significantly high *CXCL14* expression correlated with a low percentage of CD25+ Treg (Figure 6D). The correlation of *CXCL14* with a number of immune modulators indicate a likely effect on multiple components of the immune system with critical roles of CD8 and Treg cells. Altogether, these analyses support a positive correlation of CXCL14 with host anti-tumor immunity of breast cancer, particularly in TNBC subtype.

## 3. Discussion

Our studies demonstrate that high *CXCL14* levels correlate with better survivals of breast-cancer patients. *CXCL14* is decreased in TNBC, and overexpression of CXCL14 attenuates breast-cancer progression through regulating immune profiles of the tumor microenvironment in a T cell-dependent manner.

The positive correlation of *CXCL14* levels with better survival in breast-cancer patients is consistent with several publications: CXCL14 prolonged survivals in glioma [23] and in colorectal carcinoma [24]. In addition, in endometrioid ovarian cancer, tumors with TP53 wild-type highly expressed *CXCL14* compared to tumors with TP53 mutation, showing better progression-free survival [25]. However, the opposite results have also been reported, for example, elevated *CXCL14* expression in tumor specimens of stage III/IV correlated with worse OS in colorectal carcinoma [26]. *CXCL14* was highly expressed in advanced ovarian-cancer patients and correlated with poor prognosis [27]. These studies suggest that the associations between CXCL14 and patient survivals are likely context dependent and need to be carefully evaluated.

The cancer inhibitory effect by CXCL14 have been reported in various cancer types, including breast cancer. CXCL14 overexpression inhibits angiogenesis, proliferation, invasion, and migration of hepatocellular carcinoma cells [28]. Overexpression of CXCL14 attenuated xenograft tumor growth and lung metastasis of MDA-MB-231 cells [29]. Prostate cells expressing CXCL14 resulted in tumor growth inhibition [30]. Additionally, CXCL14 transgenic mice showed a suppressed rate of carcinogenesis and decreases in tumor volume and lung metastasis [31]. However, CXCL14 has been shown to be involved in osteolytic bone metastasis from lung cancer [32]. Those studies focused on the cell intrinsic function of CXCL14 following genetic manipulation of CXCL14 in cancer cells.

We uncovered an extrinsic function of CXCL14 on the immune microenvironment. CXCL14 is known to be responsible for immune cell infiltration, dendritic cell maturation, upregulation of major histocompatibility complex (MHC)-I expression, and cell mobilization [33]. M2 polarized macrophages play critical roles in metastatic progression. Studies from the 4T1 mouse model show the majority of macrophages are M2 polarized at the late stage of metastasis, which is the time point when we found significant reduction of macrophages in the lungs of mice bearing tumors with CXCL14 overexpression (Figure 3A, third panel). Thus, the decreased M2 macrophages likely contribute to metastasis suppressive function of CXCL14. Ly6G+ cells are one of key immune subsets that promotes metastatic progression in various cancer types, including breast cancer [34,35]. In the current study, we found an increased infiltration of Ly6G+ cells in primary tumors with CXCL14 overexpression at Day 28 (Appendix A). It is not clear whether these cells or subsets of these cells might have any anti-tumor function in the tumor suppressive roles of CXCL14. Further studies using single cell sequencing technology will likely provide answers. In tumor immunity, CD4+CD25+ Treg cells inhibit anti-tumor immunity and correlate with decreased patient survival [36]. Our study showed decreased CD4+CD25+ Treg with high expression of CXCL14 in single-cell dataset and the in vivo experiment, while the CD4 and CD8 T cells were increased. Thus, CXCL14 modulates the tumor microenvironment and enhances anti-tumor immunity.

In our effort to understand the molecular mechanisms of CXCL14 function, we suspected that CXCL14 may be involved in competitive binding of CXCL1/2/5/8 to the CXCR2 receptor. CXCL14 protein has sequence homology to these chemokines that signal through the CXCR2. However, there is a lack of evidence for the direct interaction of CXCL14 with CXCR2 [37,38]. Nevertheless, it is of great interest to investigate whether CXCL14 can act as an antagonist to the well-established CXCL1/5/8-CXCR2 signaling in the tumor microenvironment.

In summary, CXCL14 is decreased in breast cancer especially in the basal subtype (Figure 7). High levels of CXCL14 reduce tumor growth and metastasis (Figure 7, blue arrows). On the other hand, low levels of CXCL14 enhance tumor growth and metastasis (Figure 7, red arrows). CXCL14 modulates immune profiles with a decrease in neutrophiles or macrophages and Treg cells, as well as an increase in CD8+ T cells. CXCL14-suppressed TNBC progression in a T cell-dependent, emphasizing CXCL14-induced alteration of Treg and CD8+ T cells on its cancer inhibitory functions. Finally, high CXCL14 correlates with increased patient survival, suggesting CXCL14 as a biomarker for prognosis for breast cancer patients.

## 4. Materials and Methods

### 4.1. Data Analysis of CXCL14 Correlation with Patient Survival and Breast-Cancer Subtypes

The breast-cancer patient survival analysis was performed using publicly available datasets, TCGA-BRCA, METBRIC [39], Yau (2010) [40] and Kaplan–Meier plotter database (http://kmplot.com/analysis/index.php?p=service&cancer=breast, accessed on 19 April 2022). The CXCL14 expression levels in subtypes of breast cancer were analyzed in, TCGA-BRCA, METBRIC [39], and Yau (2010) [40] dataset.

### 4.2. Cell Line and Cell Culture

The 4T1 (ATCC^®^ CRL-2539™) cells were purchased from the American Type Culture Collection (ATCC, Manassas, VA). The 4T1 TNBC cells were cultured at 37 °C in a water-saturated atmosphere of 95% air and 5% CO_2_ with Dulbecco’s modified Eagle’s medium, supplemented with 10% heat-inactivated fetal bovine serum, 1% Antibiotic-Antimycotic, at 37 °C in a humidified atmosphere containing 5% CO_2_ and confirmed to be mycoplasma negative.

### 4.3. Generation of CXCL14 Overexpressed Cell Line

The lentiviral vector containing CXCL14 was purchased from Genecopedia (Rockville, MD, USA). The lentiviral CXCL14 vectors were constructed by inserting synthesized mouse DNA for CXCL14 (BRAK) into ampicillin resistant pFUGW lentiviral expression vector. CXCL14 overexpressed 4T1 cell clones were generated by stably transfecting the constructed CXCL14 vector, followed by puromycin selection (Appendix A). CXCL14 levels in cell lysates were measured by mouse CXCL14/BRAK enzyme-linked immunosorbent assay (ELISA) kit (Novus Biologicals, Centennial, CO, USA) according to the manufacturer’s instructions. The optical density of each well was determined using a microplate reader at 450 nm wavelength. The cell clone with the highest CXCL14 protein levels was selected and expanded for further study.

### 4.4. Single Cell Preparation from Tissue Samples

Tumor and lung tissues were treated with dissociation buffer (3 mL of DMEM + 5% FBS, 30 uL of collagenase [0.1 g/1 mL in PBS], 30 uL of Dispase/neutral protease [0.012 g/1 mL in PBS], and 30 uL of DNase [0.015 g/1 mL]) for 45 min in 37 °C at 150 rpm. The tumor- and lung-tissue homogenates were passed through a 70 μm cell strainer to obtain single cell suspensions. The single cell suspensions were washed with 20 mL of Sorting Buffer [500 mL PBS, 10 mL FBS,1 mL 0.5 M EDTA]; treated with 10 mL of ACK Lysis Buffer for 5 min on ice; resuspended with 5–10 mL Sorting Buffer. The cells were counted to 1–2 million cells (100 uL); incubated with antibodies (Appendix A) in the dark and on ice for 30 min; washed to remove the antibodies by centrifuging at 1200 rpm for 5 min; resuspended in 500 uL Sorting Buffer for flow cytometry analysis. 

### 4.5. Flow Cytometry Analysis

Single cells were stained by the panels of markers: CD45+, LY6G+, LY6C+, CD11B+ F480+, CD3+, CD4+, CD8+, CD25+, CD19+, and NK1.1+ (Appendix A, Thermo Fisher Scientific, Waltham, MA, USA), and stained with DAPI to determine cell viability. The stained cells were analyzed by flow cytometry (LSRFortessa, BD Bioscience).  We gated on CD45+7AAD- to mark live immune cells. For myeloid cells, we used CD11b marker in live CD45 population to gate on myeloid cells. Next, we used exclusion gating strategy by using CD11b+ Ly6G-Ly6C+   (Monocytic myeloid cells) and CD11b+F80   (macrophages). For lymphoid cells, we used CD3 in live CD45+ cells to gate T cells and CD19 to gate on B cells. CD3+ population were further gated to look into different T cells subtypes. Cell. subtypes, CD4 (CD4 T cell subtype). CD4+CD25+   (Treg cells) and CD8   (CD8 T cell subtype).  The data were analyzed by FlowJo (BD Bioscience).

### 4.6. In Vivo Animal Study

BALB/c mice (female, 6- to 8-week-old) were purchased from Charles River. All animal protocols are approved by the National Cancer Institute’s Animal Care and Use Committee. We injected immunocompetent BALB/c mice with 4T1 cells (3 × 10^5^ cells/0.2 mL) containing an empty vector (4T1-Control) or with the 4T1 overexpressing CXCL14 (4T1-CXCL14-OE) into the mammary fat pad #2 (*n* = 25 per group), and the tumors and the lungs were collected for further phenotypic analysis at day 12, 18, and 28. Mice were euthanized 12, 18, and 28 days later for collection of tumor and lung tissues.

The tumors were measured at the end point using a scale and calipers and the tumor size was calculated as volume = length × width^2^ × 0.5. The number of lung metastases was evaluated by directly counting of small whiter metastases after tissue fixation.

### 4.7. H&E Staining

Frozen lung sections were placed into OCT and stained for H&E staining.

### 4.8. Human Correlation of CXCL14 with the Immune Microenvironment

The purity-adjusted spearman’s correlation coefficient of CXCL14 expression with immune cell subsets, which were estimated by xCell cell type enrichment analysis, was calculated by using TIMER2.0 in the TCGA breast-cancer dataset [41] for the association of CXCL14 with immune subsets. The correlation of CXCL14 with Cytotoxic T lymphocyte levels (CTL score) in TNBC dataset (GSE58812) was analyzed by using Tumor Immune Dysfunction and Exclusion (TIDE) (http://tide.dfci.harvard.edu/, accessed on 12 May 2020). METABRIC breast-cancer dataset was downloaded with MetaGxBreast R library v1.12.0. The NIH Integrative Data Platform (NIDAP, https://nidap.nih.gov, accessed on 30 March 2022) was used to access and process TCGA breast cancer data from the Genomic Data Commons (GDC) data portal; HTseq counts were filtered for low-count and normalized with limma-voom and quantile normalization using R Bioconductor limma library v3.38.3. Molecular subtypes and TNBC classification of METABRIC and TCGA patients were adopted from Lehmann et al. [42]. Correlation analysis and visualization were performed using R programming language v3.5.1 and ggplot2 R library v3.3.0. The single cell RNA-seq data of human breast cancer (GSE107764) [43] was analyzed to investigate the correlation of CXCL14 with immune cell subsets. The immune cell annotations were extracted from Single Cell portal (https://singlecell.broadinstitute.org/single_cell/study/SCP1039/a-single-cell-and-spatially-resolved-atlas-of-human-breast-cancers#study-download, accessed on 22 April 2022). The percentages of immune cell subsets in the total immune cells were calculated and compared in the patients with high and low CXCL14 expression.

### 4.9. Statistical Analysis

GraphPad Prism v6.01 and R were used for graphs and statistics. Unless otherwise indicated, data were expressed as mean ± SD. All data were analyzed using the Student’s *t*-test for comparison of two groups or one-way ANOVA for three groups or more. Differences were considered statistically significant when the *p*-value was <0.05.

## 5. Conclusions

In conclusion, CXCL14 plays a critical role in inhibiting tumor growth and lung metastasis of TNBC in a T-cell dependent manner, through modulating the TNBC tumor immune contexture. Also, high CXCL14 correlates with better survival in patients with breast cancer, and could serve as a prognostic biomarker for clinical features.

## Figures and Tables

**Figure 1 ijms-23-09314-f001:**
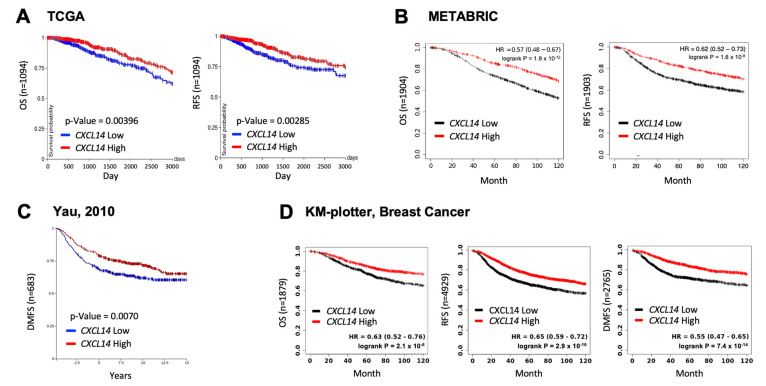
Correlation between *CXCL14* levels and survival in patients with breast cancer. (**A**) Kaplan–Meier curves for overall survival (OS) and recurrence-free survival (RFS) between CXCL14-low and -high patients in TCGA breast-cancer dataset. (**B**) Kaplan–Meier curve for OS and RFS between CXCL14-low and -high patients in MATABRIC dataset. (**C**) Kaplan–Meier curves for distant metastasis-free survival (DMFS) between CXCL14-low and -high patients in Yau 2010 dataset. (**D**) Kaplan–Meier curves for OS, RFS, and DMFS between CXCL14-low and -high patients in the Kaplan–Meier plotter database (http://kmplot.com/analysis/index.php?p=service&cancer=breast, accessed on 4 May 2022).

**Figure 2 ijms-23-09314-f002:**
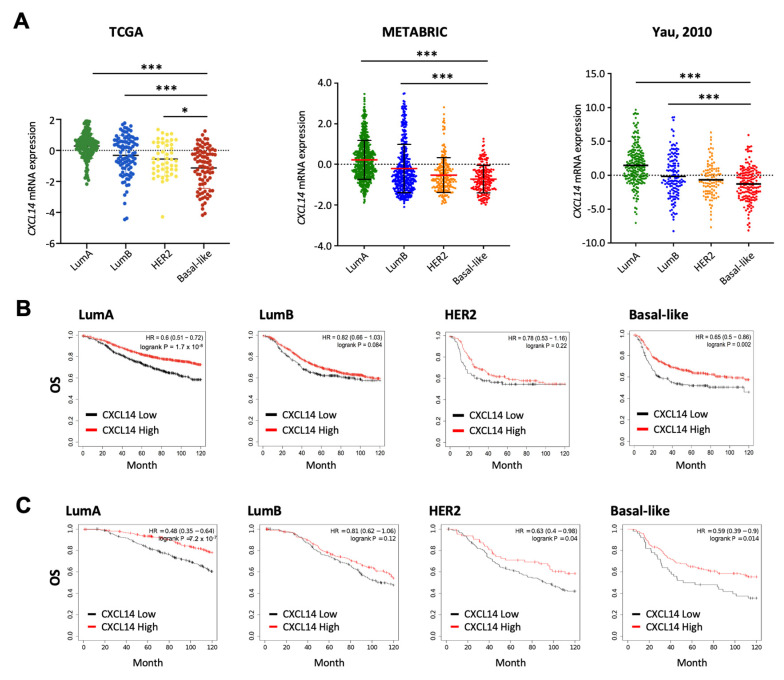
CXCL14 expression levels and its correlation with survivals in breast-cancer subtypes. (**A**) Comparison of CXCL14 expression levels between luminal A (LumA), luminal B (LumB), HER2, and basal-like breast-cancer subtypes in TCGA (*n* = 945), METABRIC (*n* = 1559), and Yau datasets (*n* = 622). (**B**,**C**) Kaplan–Meier curves for OS between CXCL14-low and -high expression levels in breast-cancer subtypes in the Kaplan–Meier plotter database (**B**) and METABRIC dataset (**C**). Data are presented as mean ± SD. * *p* < 0.05, *** *p* < 0.001.

**Figure 3 ijms-23-09314-f003:**
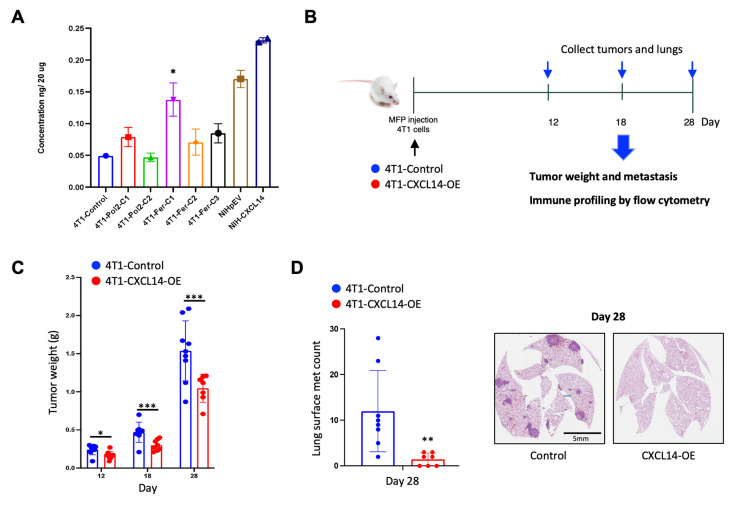
Effects of CXL14 overexpression on tumor growth and metastasis in 4T1 orthotopic model of spontaneous metastasis. (**A**) CXCL14 protein levels in generated CXCL14 overexpression (OE) 4T1 cell clones and NIH-fibroblast cells by ELISA. * indicates 4T1-CXCL14-OE cells selected for an in vivo study. (**B**) Experiment design for in vivo study to investigate the functional role of CXCL14 on TNBC progression. (**C**) Comparison of tumor weights in mice bearing 4T1-control and CXCL14-OE 4T1 cells, at day 12, 18, and 28 after cancer-cell injection (*n* = 6–8). (**D**) Lung metastases in mice bearing 4T1-control and CXCL14-OE 4T1 cells at post-injection day 28 (left panel: *n* = 7–9) and representative H&E staining images (right panel). Data are presented as mean ± SD. * *p* < 0.05, ** *p* < 0.01. *** *p* < 0.001.

**Figure 4 ijms-23-09314-f004:**
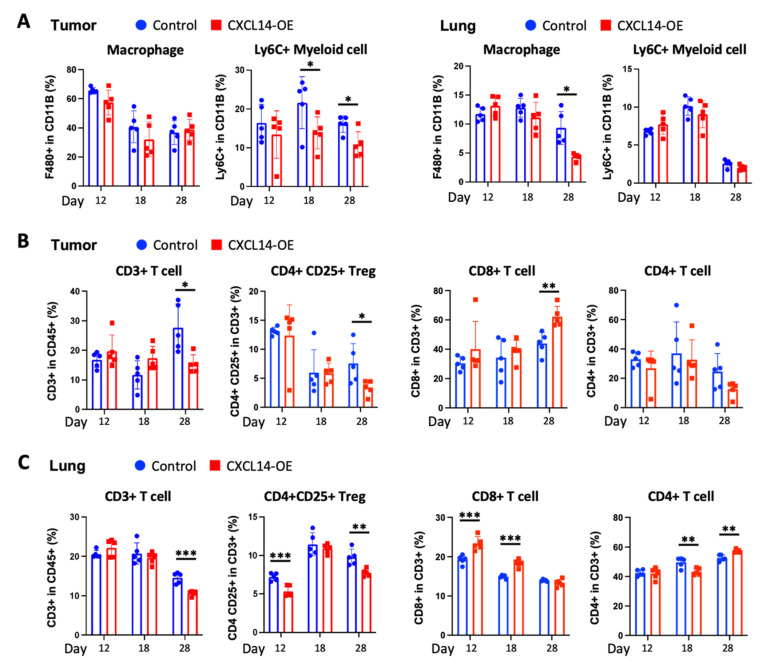
CXCL14-induced immune-cell profiles in the tumor microenvironment of primary tumors and lung metastatic sites. (**A**) FACS analysis of CXCL14-induced myeloid cell profiles in primary tumors and metastatic lungs at day 12, 18, and 28, after mammary fat pad injection of control and CXCL14-OE 4T1 cells (*n* = 5). (**B**,**C**) FACS analysis of CXCL14-induced lymphoid cell profiles in primary tumors and metastatic lungs at day 12, 18, and 28, after mammary fat pad injection of control and CXCL14-OE 4T1 cells (*n* = 5). Data are presented as mean ± SD. * *p* < 0.05, ** *p* < 0.01, *** *p* < 0.001.

**Figure 5 ijms-23-09314-f005:**
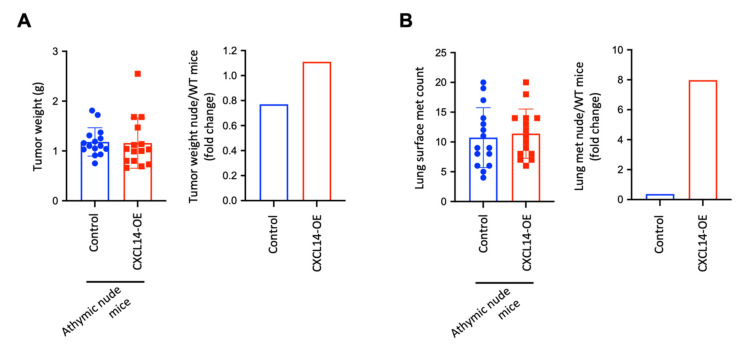
T cell-dependent effects of CXCL14 on tumor growth and metastasis. (**A**) Left panel: tumor weights of Control and CXCL14-OE 4T1 cells in T cell-deficient athymic nude mice (*n* = 15); right panel: fold changes of tumor weights in nude mice compared with WT mice. (**B**) Left panel: number of lung metastases of Control and CXCL14-OE 4T1 cells in T cell-deficient athymic nude mice (*n* = 15); right panel: fold changes of lung met counts in nude mice compared with WT mice. Data are presented as mean ± SD.

**Figure 6 ijms-23-09314-f006:**
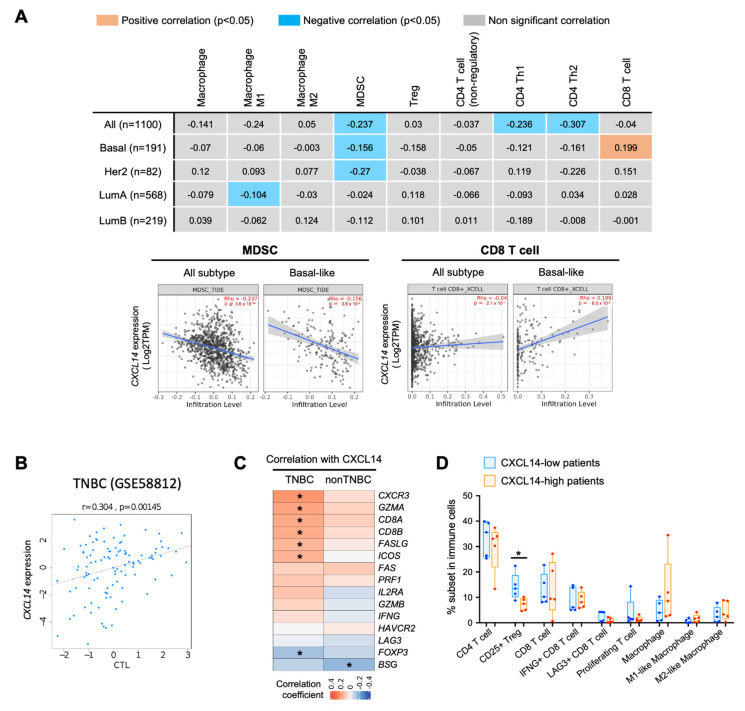
Correlation of *CXCL14* expression with immune profile. (**A**) Correlation coefficient between *CXCL14* expression and immune cell subset in TCGA breast-cancer dataset (upper panel) and scatter plots of *CXCL14* expression and immune infiltration *p* < 0.05 in Basal-like breast cancer (lower panel). (**B**) Correlation of *CXCL14* expression with cytotoxic T lymphocyte level (CTL score) by TIDE (Tumor Immune Dysfunction and Exclusion) analysis in TNBC dataset (GSE58821). (**C**) Correlation coefficient between *CXCL14* and T cell functional markers in METABRIC dataset. (**D**) Comparison of immune cell profile between patients with low- and high- CXCL14 in single cell RNA-seq data (GSE107764). Data are presented as mean ± SD. * *p* < 0.05.

**Figure 7 ijms-23-09314-f007:**
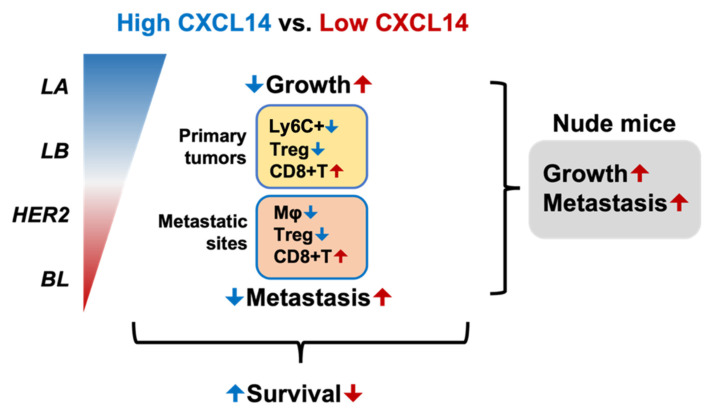
Schematic presentation of CXCL14-induced reduction of TNBC progression. Basal-like (BL) subtype including TNBC has the lowest levels of CXCL14 compared to other subtypes, such as luminal A (LA), luminal B (LB), and HER2. High levels of CXCL14 correlate with improved survival in breast-cancer patients. High levels of CXCL14 reduce tumor growth and metastasis by modulating immune profiles as follows: decreased Ly6C+ myeloid cells (Ly6C+) and Treg infiltration, and increased CD8+ T cell infiltration in primary tumors; decreased macrophages (Mφ) and Treg infiltration and increased CD8+ T cell infiltration in metastatic lungs. T cell deficient condition blocks CXCL14-induced reduction of TNBC progression. Based on these results, we concluded that CXCL14 attenuates TNBC progression by regulating immune-cell profiles of the tumor microenvironment in a T cell-dependent manner. Blue and red letters in primary tumors and metastatic sites indicate decrease and increase in immune cell infiltration, respectively.

## Data Availability

The links of data are provided at the section where data are presented.

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
