# Peer review of "CXCL14 Attenuates Triple-Negative Breast Cancer Progression by Regulating Immune Profiles of the Tumor Microenvironment in a T Cell-Dependent Manner"

_ijms, 2022, doi:10.3390/ijms23169314_

Round 1

Reviewer 1 Report

Section 2.4. The authors miscalculated the number of macrophages. They calculated the macrophage number as the number of CD45+CD11b+F4/80+ cells in the number of CD45+CD11b+ cells. If CXCL14 affects the total number of myeloid cells, this result is also associated with changes in macrophage numbers. The number of macrophages should be relative to the total number of living cells. Also Ly6c is a macrophage marker.

the number of individual groups of T cells should be in relation to all living cells.

Additionally, the authors should calculate the effect of CXCL14 on the number of M1 and M2c polarized macrophages. Macrophages can be pro-cancerous and anti-cancerous. The change in the number of macrophages says nothing about the consequences for the tumor.

The authors should also estimate the number of all types of myeloid cells, particularly neutrophils but also eosinophils and basophils.

The authors write that CXCL14 overexpression did not affect the number of B cells. In the supplement in the figure, the authors placed the * sign, which means that it does affect.

The authors should elaborate on the methodology of flow cytometer experiments, in particular cell gating. It was not stated in the main text that the number of macrophages was gated against CD45.

Reviewer 2 Report

The authors did a thorough study on investigation of the function of CXCL14 on regulating tumor microenvironment in a triple-negative breast cancer model. This article discussed the correlation between CXCL14 expression level and survival rate in patients, CXCL14 levels in different breast cancer subtypes, CXCL14’s function in inhibiting tumor growth and metastasis by regulating T cell populations, etc. Overall, the manuscript did a comprehensive investigation regarding the CXCL14 in tumor microenvironment and will attack wide interest in the field of triple-negative breast cancer therapy. Here are some comments to the authors.

1.       As CXCL14 plays an important role in inhibiting tumor growth and metastasis, the readers would be benefit from a more detailed discussion regarding existed or potential therapies utilizing CXCL14, such as targeting reagents and data in animal models.

2.       Could the authors validate the rationale of the tumor harvest time points (Day 12, 18 and 28)?

3.       Could CD4+CD25+ represent Treg cells as the authors also mentioned FOXP3 is the key functional marker for Treg?

Author Response

Reviewer #2 (Comments and Suggestions for Authors):

The authors did a thorough study on investigation of the function of CXCL14 on regulating tumor microenvironment in a triple-negative breast cancer model. This article discussed the correlation between CXCL14 expression level and survival rate in patients, CXCL14 levels in different breast cancer subtypes, CXCL14’s function in inhibiting tumor growth and metastasis by regulating T cell populations, etc. Overall, the manuscript did a comprehensive investigation regarding the CXCL14 in tumor microenvironment and will attack wide interest in the field of triple-negative breast cancer therapy. Here are some comments to the authors.

  1. As CXCL14 plays an important role in inhibiting tumor growth and metastasis, the readers would be benefit from a more detailed discussion regarding existed or potential therapies utilizing CXCL14, such as targeting reagents and data in animal models.

We appreciate the reviewer’s points. The specific receptor for CXCL14 remains uncharacterized, therefore no pharmaceutical drugs such as agonists are available for cancer treatment. However, in inflammatory disease, different from cancer setting, CXCL14 neutralizing antibodies were investigated in liver injury, stroke and skin infection in animal models [1-3].

  1. Could the authors validate the rationale of the tumor harvest time points (Day 12, 18 and 28)?

We selected the three time points based on our previous studies with the 4T1 model [4,5]. Day 12 represents pre-metastatic niche stage when inflammatory immune cells infiltrated to the distant sites prior to the arrival of cancer cells. Day 18 represents middle stage of metastasis when cancer cells were fully colonized and formed micro-metastases. Day 28 represents late stages of metastasis when a number of overt macro-metastases were formed.

  1. Could CD4+CD25+ represent Treg cells as the authors also mentioned FOXP3 is the key functional marker for Treg?

Yes, CD4+CD25+ as cell surface markers are widely used in addition to FoxP3 to identify Treg [6]. Moreover, CD25 is often used to deplete the Treg population in cancer studies [7].

References

  1. Li, J.; Gao, J.; Yan, D.; Yuan, Y.; Sah, S.; Satyal, U.; Liu, M.; Han, W.; Yu, Y. Neutralization of chemokine CXCL14 (BRAK) expression reduces CCl4 induced liver injury and steatosis in mice. Eur J Pharmacol 2011, 671, 120-127, doi:10.1016/j.ejphar.2011.09.174.
  2. Maerki, C.; Meuter, S.; Liebi, M.; Muhlemann, K.; Frederick, M.J.; Yawalkar, N.; Moser, B.; Wolf, M. Potent and broad-spectrum antimicrobial activity of CXCL14 suggests an immediate role in skin infections. J Immunol 2009, 182, 507-514, doi:10.4049/jimmunol.182.1.507.
  3. Lee, H.T.; Liu, S.P.; Lin, C.H.; Lee, S.W.; Hsu, C.Y.; Sytwu, H.K.; Hsieh, C.H.; Shyu, W.C. A Crucial Role of CXCL14 for Promoting Regulatory T Cells Activation in Stroke. Theranostics 2017, 7, 855-875, doi:10.7150/thno.17558.
  4. Yan, H.H.; Pickup, M.; Pang, Y.; Gorska, A.E.; Li, Z.; Chytil, A.; Geng, Y.; Gray, J.W.; Moses, H.L.; Yang, L. Gr-1+CD11b+ myeloid cells tip the balance of immune protection to tumor promotion in the premetastatic lung. Cancer Res 2010, 70, 6139-6149, doi:10.1158/0008-5472.CAN-10-0706.
  5. Pang, Y.; Gara, S.K.; Achyut, B.R.; Li, Z.; Yan, H.H.; Day, C.P.; Weiss, J.M.; Trinchieri, G.; Morris, J.C.; Yang, L. TGF-beta signaling in myeloid cells is required for tumor metastasis. Cancer Discov 2013, 3, 936-951, doi:10.1158/2159-8290.CD-12-0527.
  6. Togashi, Y.; Shitara, K.; Nishikawa, H. Regulatory T cells in cancer immunosuppression - implications for anticancer therapy. Nat Rev Clin Oncol 2019, 16, 356-371, doi:10.1038/s41571-019-0175-7.
  7. Fontenot, J.D.; Gavin, M.A.; Rudensky, A.Y. Foxp3 programs the development and function of CD4+CD25+ regulatory T cells. Nat Immunol 2003, 4, 330-336, doi:10.1038/ni904.

Round 2

Reviewer 1 Report

The authors did not correct the article in accordance with 1-5 comments from the reviewer.

Point 1. Authors should transfer the results from the responses to the main file of the article.

Point 2, CD45 is a leukocyte marker. The authors analyzed the number of individual groups of leukocytes into CD45 cells. If the tested chemokine increases the number of, for example, only NK cells, then, according to the analysis performed, it reduces the number of all other groups of leukocytes.

"monocyte count" / "total CD45+ cell number"

is greater than

"monocyte count" / ("CD45+ cell number from the experiment where CXCL14 expression was not altered" + "recruited NK cells")

The authors replied off topic to comments 3 and 4. reviewer's note.

Point 5. Authors should move B cell analysis results to the main article file.

Author Response

Dear Editors and Reviewers,

The authors did not correct the article in accordance with 1-5 comments from the reviewer.

Point 1. Authors should transfer the results from the responses to the main file of the article.

            For the consistency of data presentation throughout paper, we added the results from the previous response in the Supplementary Figure 3D. We wrote a description appropriately in the main text, focusing on these findings. We can move them to the main figure based on editorial preference.

Point 2, CD45 is a leukocyte marker. The authors analyzed the number of individual groups of leukocytes into CD45 cells. If the tested chemokine increases the number of, for example, only NK cells, then, according to the analysis performed, it reduces the number of all other groups of leukocytes.

"monocyte count" / "total CD45+ cell number"

is greater than

"monocyte count" / ("CD45+ cell number from the experiment where CXCL14 expression was not altered" + "recruited NK cells")

            We understand the reviewer’s question that whether the changes in CD45+ cells among the live cells contribute to the changes in immune cell subset. In our further analysis of CD45+ cells in live cells presented below, we did not observe a substantial change in CD45+ cells by CXCL14 overexpression, and the data has been added to Supplementary Figure 3E. For example, there were no significant differences at Day 18 and at Day 28, which are the time points that we found most differences in immune cell composition by CXCL14 overexpression (Figure 4). Further, in CXCL14 overexpressing tumors at Day 12, the CD45+ cells showed significant increase. However, no difference was found for the Ly6C+ myeloid cells and macrophages in the CXCL14 overexpression at Day 12.

The authors replied off topic to comments 3 and 4. reviewer's note.

Point 3. Additionally, the authors should calculate the effect of CXCL14 on the number of M1 and M2c polarized macrophages. Macrophages can be pro-cancerous and anti-cancerous. The change in the number of macrophages says nothing about the consequences for the tumor.

In the current experimental design, we did not have the markers to distinguish between M1 and M2 polarized macrophages.  Studies with 4T1 animal model show the majority of tumor associated macrophages (TAMs) are M2 polarized at the late stage of metastasis which is the time point when we found significant reduction of macrophages in the lungs of CXCL14 overexpression group (Figure 3A, third panel). We expect most of the decreased macrophages by CXCL14 overexpression should be M2 polarized ones, which can support tumor suppressive function of CXCL14 in this study. (We added this to the Discussion). Page 5, highlighted, Line 153.

Point 4. The authors should also estimate the number of all types of myeloid cells, particularly neutrophils but also eosinophils and basophils.

            We did not use eosinophil and basophil markers. The biological functions of eosinophils and basophils in TNBC are not well characterized. In addition, in the 4T1 TNBC model, the normal differentiation of the myeloid lineage is disturbed and skewed toward expansion of immune-suppressive immature myeloid cells as the tumor progressed. The roles of MDCSs in metastatic progression of 4T1 cancer cells are well established [1-3]. Therefore, we focused our investigation on the tumor associated MDSCs rather than all types of myeloid cells.

Although neutrophils and the Ly6G+ MDSCs (called granulocytic or PMN (polymorphonucelar)-MDSCs) share same surface markers, they are quite different in functions. The expansion of Ly6G+ MDSCs and its immunosuppressive function are well established in 4T1 model. Here we show a tumor inhibitory role of CD11B+ Ly6G+ cells in cancer and for early stage of tumorigenesis. We believe this can be a new mechanism that uses IL-17/Th2 pathway compared to the widely known Th1 pathway of immune suppression [4]. While there are many more papers to support pro-tumor functions of Ly6G+ MDSCs in breast cancer, we recognize this is a contradictory increase in Cd11b+ Ly6G+ cells at Day 28 in the CXCL14 overexpression. We have added this analysis with an appropriate description to the Results section in the main text. Page 5, highlighted, Line 162.

Point 5. Authors should move B cell analysis results to the main article file.

We appreciate reviewer’s suggestion. We put the B cell data in the Supplementary Figure alongside newly added NK cell data  (Supplementary Figure 3F) and provided a better description of the B cell analysis and result in the main text. Page 6, highlighted, Line 175.

In our data, we found a significant decrease of CD19+ B cells in the CXCL14 overexpression compared to the control at Day 28. We agree with the reviewer that although it remains unclear what the role of CD19+ B cells are in tumor progression, B lymphocytes have been shown to drive chronic inflammation in a murine model of inflammation-associated epithelial cancer, contributing to tumor-promoting processes such as an angiogenesis, epithelial cell proliferation, and further recruitment of immune cells [5].

References     

  1. Yan, H.H.; Pickup, M.; Pang, Y.; Gorska, A.E.; Li, Z.; Chytil, A.; Geng, Y.; Gray, J.W.; Moses, H.L.; Yang, L. Gr-1+CD11b+ myeloid cells tip the balance of immune protection to tumor promotion in the premetastatic lung. Cancer Res 2010, 70, 6139-6149, doi:10.1158/0008-5472.CAN-10-0706.
  2. Bosiljcic, M.; Cederberg, R.A.; Hamilton, M.J.; LePard, N.E.; Harbourne, B.T.; Collier, J.L.; Halvorsen, E.C.; Shi, R.; Franks, S.E.; Kim, A.Y.; et al. Targeting myeloid-derived suppressor cells in combination with primary mammary tumor resection reduces metastatic growth in the lungs. Breast Cancer Res 2019, 21, 103, doi:10.1186/s13058-019-1189-x.
  3. Ouzounova, M.; Lee, E.; Piranlioglu, R.; El Andaloussi, A.; Kolhe, R.; Demirci, M.F.; Marasco, D.; Asm, I.; Chadli, A.; Hassan, K.A.; et al. Monocytic and granulocytic myeloid derived suppressor cells differentially regulate spatiotemporal tumour plasticity during metastatic cascade. Nat Commun 2017, 8, 14979, doi:10.1038/ncomms14979.
  4. Liu, Y.; O'Leary, C.E.; Wang, L.S.; Bhatti, T.R.; Dai, N.; Kapoor, V.; Liu, P.; Mei, J.; Guo, L.; Oliver, P.M.; et al. CD11b+Ly6G+ cells inhibit tumor growth by suppressing IL-17 production at early stages of tumorigenesis. Oncoimmunology 2016, 5, e1061175, doi:10.1080/2162402X.2015.1061175.
  5. Gu, Y.; Liu, Y.; Fu, L.; Zhai, L.; Zhu, J.; Han, Y.; Jiang, Y.; Zhang, Y.; Zhang, P.; Jiang, Z.; et al. Tumor-educated B cells selectively promote breast cancer lymph node metastasis by HSPA4-targeting IgG. Nat Med 2019, 25, 312-322, doi:10.1038/s41591-018-0309-y.

Round 3

Reviewer 1 Report

ok